# A recent update on the morphological classification of intraductal papillary neoplasm of the bile duct: Correlation with postoperative prognosis and pathological features

Daisuke Noguchi[1], Naohisa Kuriyama[1]*, Yuji Kozuka[2], Haruna Komatubara[1], Tatsuya Sakamoto[1], Takahiro Ito[1], Aoi Hayasaki[1], Yusuke Iizawa[1], Takehiro Fujii[1], Akihiro Tanemura[1], Yasuhiro Murata[1], Masashi Kishiwada[1], Shugo Mizuno[1]

1 Department of Hepatobiliary Pancreatic and Transplant Surgery, Mie University Graduate School of Medicine, Japan, 2 Department of Pathology, Mie University Graduate School of Medicine, Japan

* naokun@med.mie-u.ac.jp

## Abstract

### Purpose

We proposed a novel morphological classification for intraductal papillary neoplasm of the bile duct (IPNB) and evaluated its association with postoperative prognosis.

### Methods

Forty-two IPNB patients who underwent surgical resection were classified morphologically into three types—branched (n = 10), main duct (n = 26), and mixed (n = 6)—based on preoperative imaging features indicating cystic and/or bile duct involvement. Among them, 32 patients with evaluable specimens were further categorized pathologically into Type 1 (n = 10) and Type 2 (n = 22). Patient characteristics and postoperative outcomes were analyzed.

### Results

Intraepithelial neoplasia was more frequently observed in the branched type, whereas invasive carcinoma predominated in the main duct type. In the mixed type, a half of patients involved both intra- and extrahepatic bile ducts, and this type also showed the highest incidence of residual tumor. The mixed type had the poorest 5-year postoperative survival rate (50%), compared to 90% in the main duct type and 100% in the branched type. It also exhibited the highest 5-year recurrence rate (62%). Among IPNB patients with associated invasive carcinoma, tumor infiltration beyond the bile duct wall (p < 0.001) and lymph node metastasis (p = 0.021) were significantly associated with poor prognosis, whereas the anatomical extent of the lesion (intrahepatic, extrahepatic, or both) was not. Morphological classification was significantly

**Data availability statement:** All relevant data are within the manuscript and its Supporting Information files.

**Funding:** The author(s) received no specific funding for this work.

**Competing interests:** The authors have declared that no competing interests exist.

correlated with pathological subtypes: the branched type was predominant in Type 1 (60%), while the main duct type predominated in Type 2 (64%) (p = 0.039).

## Conclusions

Our novel morphological classification of IPNB correlates with postoperative prognosis and may assist in preoperative planning of surgical strategies for IPNB patients.

## Introduction

Intraductal papillary neoplasm of the bile duct (IPNB) was defined by the World Health Organization (WHO) in 2010. The concept of IPNB continues to evolve, and in the latest WHO classification (fifth edition, 2019), it is defined as a grossly visible pre-malignant neoplasm with intraductal papillary or villous growth of biliary-type epithelium [1]. The prevalence of IPNB is rare, accounting for approximately 4–15 percent of biliary tract tumors, and its pathophysiological characteristics remains incompletely understood [2]. IPNB exhibits unique features distinct from other biliary tract tumors, including papillary growth [3] and mucin production [4], which resemble to those of pancreatic intraductal papillary mucinous neoplasm (IPMN), compared rather than a tumor in biliary tract. And thus, historically, IPNB has been discussed in contrast to pancreatic IPMN due to these similarities [5,6]. In pancreatic IPMN, morphological differences such as branched duct, main duct, and mixed types reflect the degree of malignancy, and based on those morphologies, therapeutic strategies are determined [7]. This is a unique feature among digestive tract tumors. Currently, common criteria for guiding treatment strategies in IPNB have not been established. However, considering IPNB as the counterpart of pancreatic IPMN, on treatment strategy for IPNB, it may be appropriate to refer to morphological classification rather than the TNM classification for biliary tract cancer. Based on this concept, we classified the morphology of IPNB in 2013, reporting their respective characteristics [8]. Several subsequent studies attempting morphological classification like ours have been reported [9–12], but none have yielded including ours results that could guide treatment strategies until now. This may be due to insufficient understanding of tumor characteristics of IPNB, particularly unique features of IPNB that are distinct from pancreatic IPMN such as the complexity of biliary tract anatomy and high malignancy [12].

To further elucidate the clinicopathological characteristics of IPNB and establish a global consensus, a collaborative study on IPNB has been conducted since 2016 by the Japan Biliary Association (JBA) and the Korean Association of Hepato-Biliary-Pancreatic Surgery (KAHBPS) [13]. The newly proposed pathological diagnostic criteria (Type 1 and Type 2 subtypes), based on observations by specialized pathologists, have had the greatest impact on the diagnosis of IPNB in recent years and were adopted in the latest WHO classification of digestive system tumors in 2019 [1].

This study aims to update the morphological classification of IPNB previously reported by us, incorporating the new consensus represented by the Type 1 and Type 2 pathological classifications, and validate its clinical significance in establishing a surgical strategy for IPNB.

## Methods

### Patient selection

This study protocol was approved by the medical ethics committee of Mie University Hospital (No. H2018-064) and was performed in accordance with the latest revision of the Declaration of Helsinki. Informed consent was waived due to the retrospective nature of the study. Participants were provided with the opportunity to refuse participation through an opt-out process, which was fully explained. All data were fully anonymized before access, under the supervision of the ethics committee. The data were accessed for research purposes on August 3rd, 2024.

From December 1976 to July 2024 at our institution, IPNB patients who underwent surgical resection were analyzed the perioperative characteristics of IPNB, such as age, gender, symptoms, tumor findings, surgical procedure, histology, patient prognosis, and type of recurrence, according to our new classification.

### Definition of IPNB and reevaluation of pathological subtypes 1 and 2

In this study, cases pathologically confirmed as IPNB according to the WHO classification were included. For cases diagnosed before the publication of the WHO 4th edition (2010) [14], pathological specimens had been retrospectively re-evaluated based on the WHO 4th edition criteria at the time of our previous study (published in 2013) [8] and confirmed as IPNB. Cases diagnosed after the previous study were pathologically confirmed at the time of diagnosis according to the latest WHO criteria (4th or 5th editions). Notably, the diagnostic criteria for IPNB did not substantially change between the 4th and 5th editions.

IPNB was defined as a pathologically confirmed papillary tumor, irrespective of benignity or malignancy, mucin production, or intrahepatic or extrahepatic location. The degree of dysplasia was classified as low-grade (including adenoma), high-grade (equivalent to carcinoma in situ, CIS), or invasive carcinoma [13]. Cases presenting cystic tumors with ovarian-like stroma (suspected hepatic mucinous cystic neoplasm) or flat epithelial lesions representing biliary intraepithelial neoplasia (BilIN) were excluded from the study cohort by careful pathological discrimination [15]. Superficial intraductal tumor spread was defined as mucosal extension greater than 20 mm from the edge of the main tumor [16].

The pathological subtypes, Type 1 and Type 2, were determined by additional pathological reevaluation based on the WHO 5th edition criteria [1]. This reevaluation was performed by a single board-certified pathologist with over 25 years of experience. Among the 42 IPNB cases, 10 cases were excluded from subtype analysis due to specimen deterioration (S1 Fig). As shown in S2 Fig, Type 1 IPNBs exhibited a well-organized papillary-villous structure with fine fibrovascular cores in a monotonous pattern. Neoplastic cells showed an overall orderly arrangement, with areas of low- to intermediate-grade dysplasia in addition to high-grade dysplasia. In contrast, Type 2 IPNBs demonstrated a more complex histological architecture, including irregular papillary branching and/or solid-tubular components, with neoplastic cells showing nuclear hyperchromasia, stratification, and loss of polarity.

### Proposal for a morphologic classification of IPNB: Branched, main duct, and mixed type

IPNB patients were classified into three types—branched, main duct, and mixed—according to our newly proposed morphological criteria, which are based on a combination of cystic and bile duct imaging features (Fig 1A). Fig 1B illustrates each type.

Given the close embryological relationship between the biliary tract and the pancreas [17], IPNB can be morphologically classified in a manner analogous to pancreatic IPMN, into branched, main duct, and mixed types [18]. Based on the international criteria for pancreatic IPMN [19], we first defined IPNBs with cystic lesions as the branched type, those with non-obstructive ductal dilatation as the main duct type, and those exhibiting both features as the mixed type. Based on these representative cases, cystic lesions were considered the "cystic factor," a defining feature of the branched type, while bile duct dilatation without tumor obstruction was considered the "bile duct factor," a defining feature of the main duct

**A.**

| Morphologic classification | | Bile duct factor | |
|---|---|---|---|
| | | *Each of following feature* <br><br> (a) Bile duct dilation without tumor obstruction <br> (b) Wall thickening / nodule in the bile duct (not in the cyst) | |
| | | **No** | **Yes** |
| **Cystic factor** | **No** | | **Main duct type** |
| | **Yes** | **Branched type** | **Mixed type** |

**B.**

**(a) Branched type**

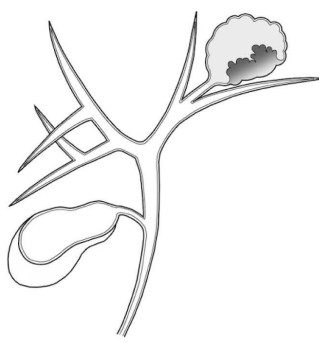

**(b) Main duct type**

Extrahepatic lesion          Intrahepatic lesion

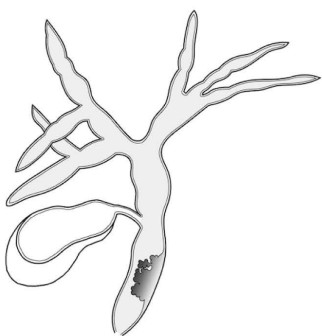 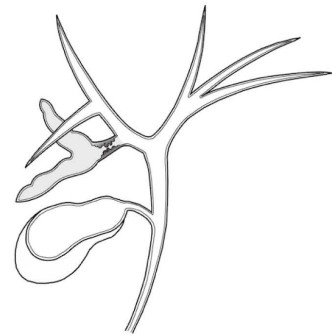

**(c) Mixed type**

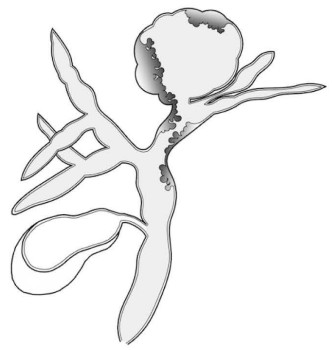

**Fig 1. Criteria for the morphologic classification of IPNB.** The IPNBs were morphologically classified into three types: branched, main duct, and mixed types, based on the specific criteria. **(A)** This criteria combine two independent factors: cystic and bile duct characteristics, which are assessed through preoperative imaging studies. **(B)** The representative schemas for each morphologic type. Abbreviations: IPNB, intraductal papillary neoplasm of the bile duct.

type. Bile duct wall thickening—sometimes accompanied by invasion into surrounding organs and mimicking cholangiocarcinoma [11]—was considered the feature to originate from large bile ducts, and thus, categorized as part of the bile duct factor. Classification criteria for the three morphological subtypes were then established based on the combination of these two factors. Representative clinical cases for each subtype are presented in S3, S4, and S5 Figs.

## Statistical analysis

Continuous variables were expressed as median and interquartile ranges and were compared using the Mann-Whitney U test for 2-group. Categorical variables were summarized as numbers and percentages and were compared using Pearson's Chi-square/ Fisher test. In the survival analysis of this study, disease-specific survival (DSS) was defined by excluding deaths such as in-hospital mortality or deaths due to other causes to evaluate the oncological prognosis of IPNB. The cumulative recurrence rate was analyzed based on the event of recurrence and the time from surgery. Survival and recurrence curves were generated using the Kaplan–Meier method, and differences between the groups were compared using the log-rank test. In statistical comparisons of perioperative variables among multiple groups, a consistent reference group was used: the branched type among morphological subtypes, intrahepatic lesions among tumor lesion extent, and invasion limited to the mucosa or fibromuscular layer among pathological tumor depth. All tests were two-sided, and p value < 0.05 was considered statistically significant. All analyses were performed using IBM SPSS Statistics version 29 (IBM Corporation, Armonk, NY).

## Results

### Clinical and pathological characteristics among three morphologic subtypes of IPNB: Branched, main duct, and mixed

Among all 42 patients with IPNB, 10 (23%) were classified as the branched type, 26 (62%) as the main duct type, and 6 (15%) as the mixed type (Table 1). Half of the patients with the mixed type experienced symptoms such as cholangitis, jaundice, or abdominal pain. Tumor lesion extent significantly differed among the three morphologic types (p < 0.001 for branched vs. main duct; p = 0.012 for branched vs. mixed). All lesions in the branched type were confined to the liver. In contrast, 69% of main duct type cases had extrahepatic tumors, and only 4% showed intrahepatic involvement. Among the mixed type, 17% had tumors in the extrahepatic bile duct, 33% in the intrahepatic duct, and 50% had lesions extending to both regions. Mural nodules were observed in 83% of all IPNB cases (n = 35), with detection rates of 90%, 77%, and 100% in the branched, main duct, and mixed types, respectively. All patients with the branched type underwent hepatectomy, with one case requiring additional extrahepatic bile duct resection. Similarly, all patients with the mixed type underwent hepatectomy, and half of them also required extrahepatic bile duct resection. In the main duct type group, 35% underwent hepatectomy, 62% underwent pancreaticoduodenectomy, and 4% underwent extrahepatic bile duct resection alone. Regarding pathological findings, low-grade intraepithelial neoplasia was more frequently observed in the branched type, whereas invasive carcinoma was more prevalent in the main duct type. Among IPNBs with associated invasive carcinoma, tumor invasion beyond the bile duct wall was observed in 0% of branched, 18% of main duct, and 25% of mixed types. In this population, the rate of residual tumor was significantly higher in the mixed type than in the branched type (75% vs. 0%, p = 0.048). Lymph node metastasis was identified in four patients (18%), all of whom had main duct type IPNB.

**Table 1. Patient demographics according to morphologic classification.**

| Variables | Branched (n = 10) | Main duct (n = 26) | Mixed (n = 6) |
|---|---|---|---|
| Age (y.o.) | 72 [66–75] | 73 [66–76] | 72 [60–77] |
| Gender (Male/ Female) | 4 (40) / 6 (60) | 17 (65) / 9 (35) | 4 (67) / 2 (33) |
| Preoperative biliary drainage (%) | 0 (0) | 2 (8) | 1 (17) |
| Symptoms (%) | | | |
| Cholangitis | 1 (10) | 4 (15) | 3 (50) |
| Jaundice | 0 (0) | 4 (15) | 3 (50) |
| Fever | 1 (10) | 1 (4) | 1 (17) |
| Abdominal pain | 0 (0) | 1 (4) | 3 (50) |
| Elevation of hepatobiliary enzyme | 1 (10) | 6 (23) | 2 (33) |
| Tumor findings by preoperative image study (%) | | | |
| Lesion extent§: Intrahepatic/ Extrahepatic/ Both | 10 (100) / 0 (0) / 0 (0) | 1 (4) / 18 (69) / 7 (27) | 2 (33) / 1 (17) / 3 (50) |
| Cyst | 10 (100) | 0 (0) | 6 (100) |
| Bile duct dilation without obstruction | 0 (0) | 3 (12) | 5 (83) |
| Bile duct wall thickness | 0 (0) | 26 (100) | 3 (50) |
| Mural nodule | 9 (90) | 20 (77) | 6 (100) |
| Lymph node swelling by image study | 0 (0) | 3 (12) | 0 (0) |
| Operative procedure (%) | | | |
| Hepatectomy | 9 (90) | 1 (4) | 3 (50) |
| Hepatectomy with extra bile duct resection | 1 (10) | 8 (31) | 3 (50) |
| Pancreaticoduodenectomy | 0 (0) | 16 (62) | 0 (0) |
| Extra bile duct resection alone | 0 (0) | 1 (4) | 0 (0) |
| Pathological findings (%) | | | |
| Low grade intraepithelial neoplasia | 5 (50) | 1 (4) | 1 (17) |
| High grade intraepithelial neoplasia (equivalent to CIS) | 0 (0) | 3 (12) | 1 (17) |
| Invasive carcinoma | 5 (50) | 22 (85) | 4 (67) |
| Invasion depth: M/ FM/ SS/ SE or SI | 4 (80) / 1 (20) / 0 (0) / 0 (0) | 5 (23) / 6 (27) / 5 (23) / 6 (27) | 2 (50) / 1 (25) / 0 (0) / 1 (25) |
| Superficial extension | 1 (20) | 9 (41) | 2 (50) |
| Lymph node metastasis | 0 (0) | 4 (18) | 0 (0) |
| Residual tumor of R1 or 2† | 0 (0) | 4 (18) | 3 (75) |
| Pathological subtypes* (%) | n = 10 | n = 16 | n = 6 |
| Type 1/ Type 2 | 6 (60) / 4 (40) | 2 (12) / 14 (88) | 2 (33) / 4 (67) |

Abbreviations: CIS, carcinoma in situ; IPNB, intraductal papillary neoplasm of the bile duct

§Tumor lesion extent significantly differed among the three morphologic types: p < 0.001 for branched vs. main duct; p = 0.012 for branched vs. mixed.

†The incidence of residual tumor was highest in the mixed type and was significantly higher than that in the branched type (p = 0.048).

*Thirty-two patients with evaluable specimens were diagnosed with pathological subtypes.

## Association between the morphologic classification and the pathologic subtypes of Type 1 and 2

32 patients without evaluable pathological specimens for the pathological classification were divided into Type 1 (n = 10) and Type 2 (n = 22) (S1 Fig). As shown in S1 Table, general background characteristics such as age, gender, and symptoms were comparable between Type 1 and Type 2. In Type 1, 70% of lesions were located intrahepatically, while only 27% of Type 2 lesions were intrahepatic (40% were in the extrahepatic duct) (p = 0.072). Over 80% of Type 2 lesions exhibited invasive carcinoma, compared to 30% in Type 1 (p = 0.007). Analyzing the association between morphological and pathological subtypes, 60% of pathological Type 1 IPNB were categorized as the branched type, 20% as the main

duct type, and 20% as the mixed type. In pathological Type 2 IPNB, the predominant morphological type was the main duct type (64%), with the branched type and mixed type each accounting for 18% (p = 0.039).

## Postoperative survival by morphology and lesion extent in all IPNB cases

In postoperative DSS in patients with the branched type IPNB, there were no postoperative deaths. The 1-, 3-, and 5-year DSS rates for patients with the main duct type were 96%, 90%, and 90%, respectively. No significant difference was observed in DSS between the branched and main duct types. However, the mixed type exhibited significantly lower 1-, 3-, and 5-year DSS rates of 100%, 50%, and 50%, respectively, compared to the branched type (p = 0.044, Fig 2A). As

**A. Morphologic classification: Branched, Main duct, or Mixed types**

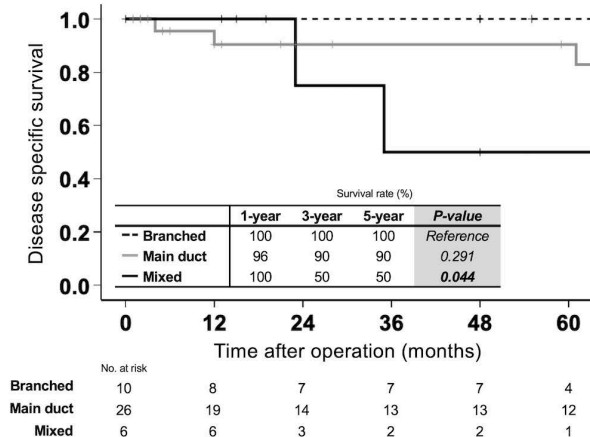

**B. Tumor lesion extent: Intrahepatic, Extrahepatic, or Both**

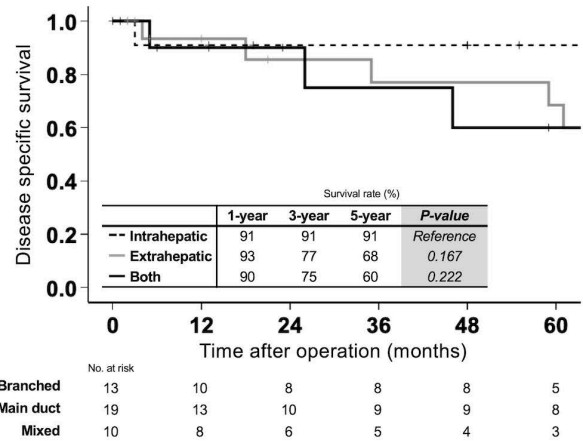

**Fig 2. Postoperative survival by morphology and lesion extent in all IPNB cases.** DSS in patients with IPNB was compared according to (A) morphological subtype—branched, main duct, and mixed—and (B) anatomical extent of the tumor—confined to the intrahepatic bile duct, extrahepatic bile duct, or both. No disease-specific deaths occurred in the branched type. There was no significant difference in DSS between the branched and main duct types. In contrast, the mixed type showed significantly worse 1-, 3-, and 5-year DSS compared to the branched type (p = 0.044). DSS did not significantly differ among the groups based on tumor extent. Abbreviations: DSS, disease-specific survival; IPNB, intraductal papillary neoplasm of the bile duct.

shown in Fig 2B, DSS did not show significant differences among the three groups based on tumor lesion extent (p = 0.167 between intrahepatic and extrahepatic bile duct lesions, p = 0.222 between intrahepatic and both lesions).

### Postoperative survival by pathological tumor progression factors in IPNB with associated invasive carcinoma

In thirty-one patients with IPNB with associated invasive carcinoma, DSS was compared based on pathological tumor progression factors. All patients with tumors confined to the mucosal (M) or fibromuscular (FM) layers survived without disease-specific death for 5 years after surgery. In contrast, patients with tumor invasion beyond the bile duct wall (serosal exposure [SE] or serosal invasion [SI]) had 1-, 3-, and 5-year DSS rates of 83%, 0%, and 0%, respectively, which were significantly lower (p < 0.001; Fig 3A). Patients with tumor extension to the subserosal (SS) or serosal (S) layers also had lower DSS compared to those with M or FM invasion (p = 0.037), although the difference became more evident after 5 years postoperatively. In patients with lymph node metastasis, the 1-, 3-, and 5-year DSS were significantly lower compared to those without lymph node metastasis. The DSS rates were 96%, 91%, and 91% versus 100%, 67%, and 67%, respectively (p = 0.021, Fig 3B). In patients with residual tumor (R1 or R2), the 1-, 3-, and 5-year DSS were 86%, 64%, and 64%, respectively. Although these rates were lower than those in patients who underwent curative resection, the difference was not statistically significant (p = 0.119, Fig 3C).

### Postoperative recurrence by morphology in IPNB with associated invasive carcinoma

The association between postoperative recurrence and morphological subtypes was evaluated in patients with invasive IPNB. No recurrence was observed in the branched type. In contrast, the mixed type showed a high recurrence incidence, with 1-, 3-, and 5-year cumulative recurrence rates of 25%, 25%, and 62%, respectively (p = 0.074 vs. branched type, Fig 4). In the main duct type, the recurrence risk was intermediate, with 1-, 3-, and 5-year cumulative recurrence rates of 6%, 6%, and 13%, respectively, and no significant difference compared to the branched type (p = 0.246).

## Discussion

The clinical significance of our proposed morphological classification for IPNB lies in three key aspects: (i) preoperative diagnostic capability, (ii) prognostic relevance based on morphological differences, and (iii) simplicity and practicality of the criteria. Given the need for straightforward application in clinical settings, our classification—based solely on cystic and bile duct factors—is highly practical (Fig 1). Notably, it is the first to demonstrate a relationship with pathological classification.

Previous morphological classifications of IPNB considered the lesion location (intrahepatic vs. extrahepatic) as a key factor [8,10,12]. In contrast, our new morphological criteria do not take lesion location into account. This is based on the understanding that IPNB can arise not only from biliary epithelial cells but also from peribiliary glands (PBGs) [20]—accessory glands distributed along both intrahepatic and extrahepatic bile ducts, which contribute to secretion and epithelial repair [18,21]. Lesions derived from biliary epithelium, when arising in small bile ducts (after the segmental branches), typically may present as the branched type and be in the intrahepatic bile ducts. In contrast, PBG-derived lesions may occur irrespective of intrahepatic or extrahepatic bile duct, as PBGs are present throughout the biliary tract. This location-independent approach allowed our classification to include atypical cases that present solely with mural nodules or wall thickening in the bile duct, without cystic changes or non-obstructive ductal dilatation. Previously, such lesions were classified differently based on their anatomical location rather than their morphological features. We believe that these lesions are fundamentally similar in morphology. Therefore, by classifying them into the main duct type based solely on the presence of the bile duct factor, regardless of lesion location, our new criteria resolve this inconsistency (S4 FigB).

Since IPNB is a precancerous lesion that can progress to invasive cancer [22], early detection and surgical resection are essential. However, a definitive surgical strategy has yet to be established. Drawing inspiration from the association between morphology and prognosis in pancreatic IPMN [19], we investigated whether our morphological classification

## A. Tumor depth

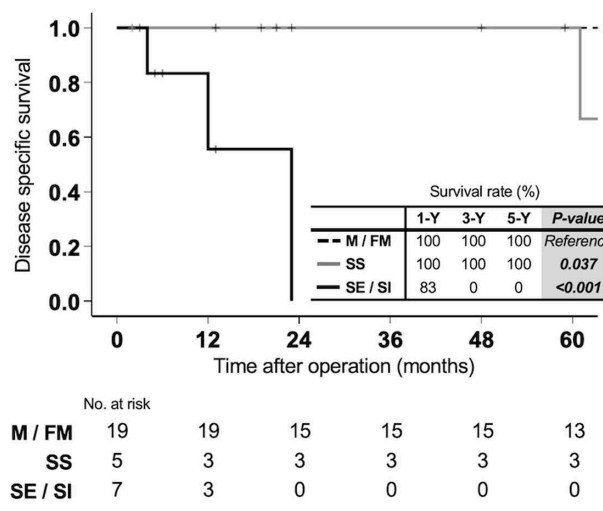

| | Survival rate (%) | | | |
|---|---|---|---|---|
| | 1-Y | 3-Y | 5-Y | *P-value* |
| -- M / FM | 100 | 100 | 100 | *Reference* |
| — SS | 100 | 100 | 100 | *0.037* |
| — SE / SI | 83 | 0 | 0 | *<0.001* |

No. at risk

| | | | | | | |
|---|---|---|---|---|---|---|
| M / FM | 19 | 19 | 15 | 15 | 15 | 13 |
| SS | 5 | 3 | 3 | 3 | 3 | 3 |
| SE / SI | 7 | 3 | 0 | 0 | 0 | 0 |

## B. Lymph node metastasis

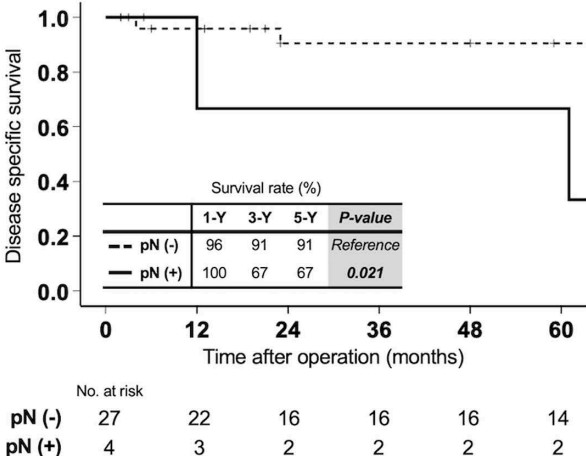

| | Survival rate (%) | | | |
|---|---|---|---|---|
| | 1-Y | 3-Y | 5-Y | *P-value* |
| -- pN (-) | 96 | 91 | 91 | *Reference* |
| — pN (+) | 100 | 67 | 67 | *0.021* |

No. at risk

| | | | | | | |
|---|---|---|---|---|---|---|
| pN (-) | 27 | 22 | 16 | 16 | 16 | 14 |
| pN (+) | 4 | 3 | 2 | 2 | 2 | 2 |

## C. Residual tumor

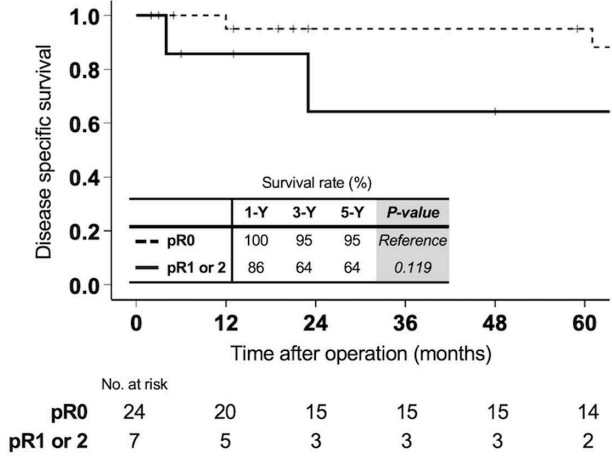

| | Survival rate (%) | | | |
|---|---|---|---|---|
| | 1-Y | 3-Y | 5-Y | *P-value* |
| -- pR0 | 100 | 95 | 95 | *Reference* |
| — pR1 or 2 | 86 | 64 | 64 | *0.119* |

No. at risk

| | | | | | | |
|---|---|---|---|---|---|---|
| pR0 | 24 | 20 | 15 | 15 | 15 | 14 |
| pR1 or 2 | 7 | 5 | 3 | 3 | 3 | 2 |

**Fig 3. Postoperative survival by pathological tumor progression factors in IPNB with associated invasive carcinoma.** In patients with invasive IPNB, DSS was analyzed according to pathological indicators of tumor progression: (A) invasion depth, (B) lymph node metastasis, and (C) residual tumor status. Patients with tumor invasion beyond the bile duct wall (SE or SI) had a 5-year DSS rate of 0%, significantly lower than those with M or FM invasion (p<0.001). The presence of lymph node metastasis was also associated with significantly worse 5-year DSS (p=0.021). Patients with residual tumor (R1 or R2) had lower 5-year DSS than those with curative (R0) resection, although this difference did not reach statistical significance (p=0.119). Abbreviations: DSS, disease-specific survival; IPNB, intraductal papillary neoplasm of the bile duct Tumor depth: M (mucosal)/ FM (fibromuscular)/ SS (subserosa)/ SE (serosal exposure)/ SI (serosal invasion).

could similarly predict the prognosis of IPNB. Among previous studies that attempted to classify IPNB morphology [9–12,23,24], ours is the first to successfully demonstrate a correlation between morphological subtype and postoperative prognosis. While Luvira et al. [9] stratified prognosis, their study excluded IPNBs in the extrahepatic bile duct—a major limitation. Among our classification—branched, main duct, and mixed—the mixed type had the poorest survival outcomes (Fig 2A). One possible explanation is the higher risk of residual tumor in the mixed type. Due to the extensive spread of lesions—half of which involved both intrahepatic and extrahepatic bile ducts—achieving R0 resection is often difficult

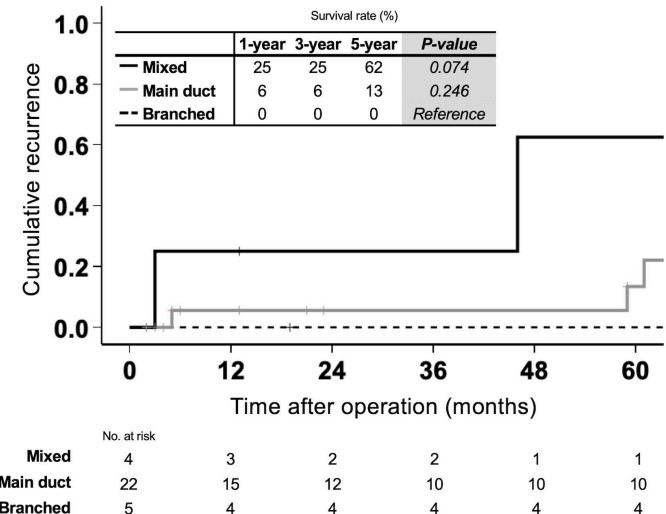

**Fig 4. Postoperative recurrence by morphology in IPNB with associated invasive carcinoma.** The association between postoperative recurrence and morphological subtypes was evaluated in patients with invasive IPNB. No recurrence was observed in the branched type. In contrast, the mixed type showed a high 5-year recurrence rate of 62%. The main duct type had an intermediate risk, with a 5-year cumulative recurrence rate of 13%. Abbreviations: IPNB, intraductal papillary neoplasm of the bile duct.

(residual tumor rate significantly higher than in the branched type, p = 0.048; Table 1). However, lesion extent alone does not appear to fully account for the poor prognosis associated with the mixed type. As shown in Fig 2B, there were no significant differences in DSS among patients with intrahepatic, extrahepatic, or both intra- and extrahepatic tumor involvement. This suggests that our classification reflects not only the anatomical extent of the tumor but also underlying biological characteristics—for example, its association with pathological subtypes. We next examined the relationship between prognosis and cancer staging. In invasive IPNB, tumor stage is determined according to different classification systems based on tumor location—namely, intrahepatic, perihilar, or distal bile duct cancer—making direct comparison across cases impractical. Instead, we focused on two common oncologic factors: depth of invasion and lymph node metastasis, which can serve as surrogates for pathological staging. As shown in Fig 3, both were strongly associated with poor prognosis and, in terms of p-values, were even more predictive than morphological subtype. However, as these factors are only assessable postoperatively, our morphological classification offers a major advantage by enabling preoperative prognostic assessment and supporting treatment strategy.

We propose several practical implications of our morphological classification for surgeons. While the choice of surgical procedure (e.g., hepatic resection or pancreaticoduodenectomy) primarily depends on the anatomical location and extent of the tumor, our classification can aid in determining the appropriate resection extent. It serves as a rationale for surgical aggressiveness when considering highly invasive procedures such as hepatopancreatoduodenectomy. For example, mixed-type IPNB, associated with poor prognosis, may require wider surgical margins. In contrast, branched-type IPNB, which typically exhibits limited invasion and lower malignancy, may allow for more limited resection and is often amenable to minimally invasive approaches such as robotic or laparoscopic surgery. Postoperatively, the classification may also inform the need for adjuvant therapy and surveillance. The mixed type demonstrated a significantly higher recurrence rate, with over half of the patients experiencing recurrence within five years (Fig 4). In contrast, no recurrence was observed in the branched type, even when invasive components were present. These findings suggest that mixed-type IPNB may benefit from adjuvant therapy and intensive follow-up, while such measures might be unnecessary in the branched type. Crucially, this classification provides prognostic information preoperatively, supporting personalized treatment planning

from the outset. Surveillance strategies like pancreatic IPMN are not recommended, even for the branched type. We encountered a case initially classified as branched type that progressed over 14 years to mixed type with diffuse bile duct involvement (S7 and S8 Figs). Given the difficulty of curative resection in advanced mixed-type IPNB, timely surgery is advisable.

To objectively evaluate the still poorly understood nature of IPNB, the pathological subtypes, Type 1 and Type 2, were proposed [13]. Some differences between these subtypes have been reported, with Type 2 showing higher malignancy (over 80% of Type 2 exhibiting invasive carcinoma compared to 30% in Type 1, p = 0.007, S1 Table). Many previous studies have reported worse postoperative survival in Type 2 IPNB compared to Type 1 [25–28]. Although our results did not show statistical significance, the disease-specific survival in Type 2 was lower than in Type 1, with a 5-year survival rate of 100% for Type 1 versus 76% for Type 2 (S6 Fig). While this pathological classification should be more critical in future clinical settings as a basis, it has limitations for preoperative management since the types can only be determined postoperatively. Our novel morphological classification is, to the best of our knowledge, the first tool to demonstrate a correlation between morphological and pathological features. In our cohort, Type 1 exhibited more branched type morphology than Type 2 (60% vs. 18%), while Type 2 exhibited more main duct type morphology than Type 1 (64% vs. 20%) (p = 0.039, S1 Table). Our morphological classification may complement the pathological classification by predicting its subtype preoperatively.

Limitations of this study include its retrospective, single-center design; however, the most critical limitation is the small cohort size. While we demonstrated an association between our morphological classification and patient prognosis, the small number of cases—particularly the limited number of mixed-type IPNBs with poor prognosis (n = 6)—precludes definitive conclusions about its generalizability. For instance, to establish the classification as an independent prognostic factor, it would be necessary to compare it with known pathological risk factors such as depth of invasion and lymph node metastasis using multivariate analysis, which is not feasible given the current sample size. Nonetheless, we believe that the concept and criteria of our novel morphological classification are of significant value and hold considerable potential in the clinical setting. Future multicenter studies with larger cohorts are needed to objectively validate the significance and reliability of this classification system.

In conclusion, we classified IPNB into three morphological types—branched, main duct, and mixed—drawing inspiration from the classification of pancreatic IPMN. This classification demonstrated a clear correlation with postoperative prognosis and may help guide surgical strategies tailored to each morphological type in patients with IPNB.

## Supporting information

**S1 Table. Patient demographics according to pathological subtype.** Abbreviations: CIS, carcinoma in situ; IPNB, intraductal papillary neoplasm of the bile duct.
(DOCX)

**S1 Fig. Study flow.** Abbreviation: BilIN, Biliary intraepithelial neoplasia; IPNB, intraductal papillary neoplasm of the bile duct.
(TIFF)

**S2 Fig. Histopathological features of the pathological subtypes of IPNB: Type 1 or 2.** (A) Pathological Type 1 of IPNB. (A-a) A monotonous pattern of well-organized papillary-villous growth with a fine fibrovascular core appears within the cyst. Neoplastic cells show an overall well-organized arrangement. (A-b) A component of low to intermediate dysplasia is present. (B) Pathological Type 2 of IPNB. (B-a) In the dilated bile duct, various papillary architectures with thick fibrovascular stalks exhibit complex, irregular, and heterogeneous growth. (B-b) Neoplastic cells display nuclear hyperchromasia, stratification, and disordered polarity. Most cells show high-grade dysplasia. Abbreviation: IPNB, intraductal papillary neoplasm of the bile duct.
(TIFF) (TIFF)

**S3 Fig. Typical case of the branched-type IPNB.** (A) CT imaging revealed a cystic lesion between liver segments 3 and 4 (arrowheads), with an enhanced mural nodule present within the cyst (arrow). (B) Endoscopic demonstrated mural nodules that were enhanced with a contrast agent (arrow). (C) ERC showed no bile duct dilation or extracystic lesions. (D) Histological examination identified a papillary growth tumor within the cystic lesion, leading to a diagnosis of IPNB associated with invasive carcinoma. Abbreviations: EUS, endoscopic retrograde cholangiography; IPNB, intraductal papillary neoplasm of the bile duct; US, ultrasound.
(TIFF)

**S4 Fig. Typical case of the main duct-type IPNB.** (A) A main duct-type IPNB with a mural nodule in the extrahepatic bile duct. (A-a) A mural nodule (arrowhead) was detected in the intrapancreatic bile duct on CT. (A-b) ERC demonstrated diffuse dilation of the entire bile duct proximal to the tumor. (A-c) The resected specimen obtained by pancreaticoduodenectomy showed a solid white tumor (*). (A-d) Histologically, the tumor exhibited a papillary architecture with heterogeneous growth patterns and was diagnosed as high-grade intraepithelial neoplasia (equivalent to CIS). (B) A main duct-type IPNB with localized wall thickening in the intrahepatic bile duct. (B-a) CT revealed wall thickening (arrowhead) and localized bile duct dilation (arrow) in the posterior branch. (B-b) PET-CT demonstrated FDG uptake in the lesion along the dilated posterior branch (arrow). (B-c) ERC showed wall thickening (arrowheads) in the corresponding area. (B-d) In the resected specimen, thickening of the posterior bile duct wall and a papillary tumor with mucin production were observed. (B-e) Histologically, the lesion consisted of papillary tumor cells, including invasive adenocarcinoma with heterogeneous growth patterns, leading to the diagnosis of IPNB with associated invasive carcinoma. Abbreviations: CIS, carcinoma in situ; CT, computed tomography; ERC, endoscopic retrograde cholangiography; FDG, fluorodeoxyglucose; PET, positron emission tomography; IPNB, intraductal papillary neoplasm of the bile duct.
(TIFF)

**S5 Fig. Typical case of the mixed-type IPNB.** The mixed type IPNB is characterized by both cystic and bile duct features. (A) CT showed a multilobular cystic lesion (arrowheads) extensively occupying in the left lobe. (B) In MRCP, entire bile duct dilation without obstruction was identified besides the intrahepatic cystic lesion (arrowheads). (C) ERC showed defects in biliary truct and a bulging papilla of Vater, resulting from hypersecretion of mucin. (D) Resected specimen of the left lobe exhibited a multilobular cystic lesion with diagnosis of IPNB with associated invasive carcinoma. Abbreviations: CT, computed tomography; ERC, endoscopic retrograde cholangiography; MRCP, magnetic resonance cholangiopancreatography; IPNB, intraductal papillary neoplasm of the bile duct.
(TIFF)

**S6 Fig. Postoperative survival by the pathological subtypes in all IPNB cases.** Postoperative DSS in patients with IPNB was compared according to pathological subtypes. Although the 1-, 3-, and 5-year survival rates in Type 2 (95%, 76%, and 76%, respectively) were lower than those in Type 1, the difference was not statistically significant (p = 0.151). Abbreviations: DSS, disease-specific survival; IPNB, intraductal papillary neoplasm of the bile duct.
(TIFF)

**S7 Fig. A 14-year follow-up case of mixed type: At the time of diagnosis.** A 57-year-old female. (A) An intrahepatic cyst (arrowhead) was incidentally detected on CT. The lesion was in segments 5–8 and exhibited a mural nodule (yellow arrow). (B) Abdominal ultrasonography revealed an enhanced mural nodule (yellow arrow) within the cystic lesion (arrowhead), and a biopsy was performed on the nodule. (C) The biopsy specimen showed no evidence of malignancy. Since there were no abnormal features in the bile duct, such as dilation or wall thickening, she was diagnosed with branched-type IPNB. Although surgery was planned, the patient chose surveillance due to the absence of malignant findings. Abbreviations: IPNB, intraductal papillary neoplasm of the bile duct.
(TIFF)

**S8 Fig. A 14-year follow-up case of mixed type: At the time of surgery.** (A) (B) Fourteen years later, the cystic lesion (arrowhead) and the mural nodule (yellow arrow) had enlarged on CT. Additionally, dilatation of both the intrahepatic bile duct (blue arrowhead) and the extrahepatic bile duct (*) was observed. (C) Cholangiography revealed a cystic lesion (arrowhead) connected to the bile duct and dilated bile ducts without obstruction (*). (D) Cholangioscopy demonstrated mucin secretion and an intraluminally growing papillary tumor in the right main hepatic duct. Morphologically, the lesion was diagnosed as the mixed-type IPNB due to the presence of both the cystic lesion and bile duct dilatation without obstruction. Right hepatectomy and caudate lobectomy with extrahepatic bile duct resection were performed. (E) In the specimen, a mucin-producing papillary tumor (yellow arrow) was identified within the cystic lesion (arrowhead). While most of the cyst wall epithelium was adenoma, the tumor was an adenocarcinoma infiltrating the fibromuscular layer. Abbreviations: IPNB, intraductal papillary neoplasm of the bile duct.
(TIFF)

**S1 Data. Dataset of resected IPNB cases.**
(XLSX)

**S2 Data. Dataset used for pathological subtype analysis.**
(XLSX)

## Acknowledgments

Not applicable.

## Author contributions

**Conceptualization:** Daisuke Noguchi, Naohisa Kuriyama, Yuji Kozuka, Shugo Mizuno.

**Data curation:** Daisuke Noguchi, Yuji Kozuka, Haruna Komatubara, Tatsuya Sakamoto, Takahiro Ito, Aoi Hayasaki.

**Formal analysis:** Daisuke Noguchi, Yusuke Iizawa, Takehiro Fujii, Akihiro Tanemura, Yasuhiro Murata, Masashi Kishiwada.

**Investigation:** Daisuke Noguchi, Yusuke Iizawa, Takehiro Fujii, Akihiro Tanemura, Yasuhiro Murata, Masashi Kishiwada.

**Methodology:** Daisuke Noguchi, Yuji Kozuka.

**Project administration:** Daisuke Noguchi, Naohisa Kuriyama, Shugo Mizuno.

**Supervision:** Naohisa Kuriyama, Shugo Mizuno.

**Validation:** Daisuke Noguchi.

**Visualization:** Daisuke Noguchi, Yuji Kozuka.

**Writing – original draft:** Daisuke Noguchi.

**Writing – review & editing:** Daisuke Noguchi, Naohisa Kuriyama, Yuji Kozuka, Haruna Komatubara, Tatsuya Sakamoto, Takahiro Ito, Aoi Hayasaki, Yusuke Iizawa, Takehiro Fujii, Akihiro Tanemura, Yasuhiro Murata, Masashi Kishiwada, Shugo Mizuno.

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
