## [Decision Letter · Decision Letter 0]

17 Mar 2025

PONE-D-25-06283A Recent Update on the Morphological Classification of Intraductal Papillary Neoplasm of the Bile Duct: Correlation with Postoperative Prognosis and Pathological FeaturesPLOS ONE

Dear Dr. Kuriyama,

Thank you for submitting your manuscript to PLOS ONE. After careful consideration, we feel that it has merit but does not fully meet PLOS ONE’s publication criteria as it currently stands. Therefore, we invite you to submit a revised version of the manuscript that addresses the points raised during the review process.

We look forward to receiving your revised manuscript.

Kind regards,

Sean Bennett, MD, MSc

Academic Editor

PLOS ONE

Journal Requirements:

Reviewers' comments:

Reviewer's Responses to Questions

**Comments to the Author**

1. Is the manuscript technically sound, and do the data support the conclusions?

Reviewer #1: Partly

Reviewer #2: Partly

2. Has the statistical analysis been performed appropriately and rigorously? 

Reviewer #1: I Don't Know

Reviewer #2: Yes

3. Have the authors made all data underlying the findings in their manuscript fully available?

Reviewer #1: Yes

Reviewer #2: No

4. Is the manuscript presented in an intelligible fashion and written in standard English?

Reviewer #1: Yes

Reviewer #2: Yes

5. Review Comments to the Author

Reviewer #1: The manuscript “A Recent Update on the Morphological Classification of Intraductal Papillary Neoplasm of the Bile Duct: Correlation with Postoperative Prognosis and Pathological Features” by Noguchi et al. examines 42 IPNBs and classifies them using a morphological system derived through radiologic characterization of IPNBs as branched type, main duct type, or mixed. They also classified them using the Japanese-Korean / WHO system as type 1 or type 2 based on histopathology. The authors evaluated post-operative survival based on several clinical-pathologic parameters, including by their proposed morphologic system. They found that mixed type IPNBs had worse survival compared to the others.

This paper represents an update on an earlier publication by this institution from 2013 and it includes more cases now. However, some fundamental flaws remain that make me skeptical of the validity of their proposed classification.

1) The mixed type only had 6 cases, a very small sample size that barely resulted in statistical differences on survival analysis. With so few cases, it is impossible to do meaningful multivariate regressions. My concern is that the mixed type may simply represent a more advanced stage disease, whereby tumour has expanded beyond its main duct or branch duct origin to involve adjacent regions. Although size or span can be difficult to measure in these types of lesions, I wonder if these are simply “larger” or at least more advanced stage tumours. In other words, I’m skeptical that the mixed morphology is an independent prognosticator that couldn’t be supplanted by other related, existing parameters such as size or stage.

2) There is some overlap between the intrahepatic versus extrahepatic anatomic classification as well as the type 1 and type 2 pathologic classifications with the proposed morphologic classification. Kim et al (reference 12) showed that intrahepatic versus extrahepatic location did not correlate with survival in a larger sample size. The authors argue that their morphologic system does correlate with survival due to more extensive spread in the mixed type. Couldn’t a similar classification be made for “mixed intra-hepatic and extrahepatic involvement” instead? Do mixed intra and extra-hepatic tumours have different prognosis from those only involving intra or only involving extra?

3) The authors posit that the branch duct tumours are originating from peri-biliary glands, but since all the branch ducts tumours were intrahepatic, couldn’t they just be from smaller branches of intrahepatic ducts, rather than peri-biliary glands?

4) With respect to survival analysis, disease specific survival is defined as per usual. However, the most common definition of “recurrence free survival” includes all deaths, but the authors chose to use a “recurrence-free disease-specific survival”, which is not a common oncologic outcome measure. I wonder if this very specific definition was chosen preferentially after multiple survival parameters were analyzed.

5) Although the authors argue that the more aggressive nature of the mixed type could guide surgical management, the bulk of surgical decision making is still based on anatomical location, which dictates whether hepatectomy, bile duct resection, or pancreaticoduodenectomy is performed.

In summary, although the finding that a mixed main/branched duct morphology potentially signals worse prognosis, this study has two main limitations. One is that the sample size for this category is very small (n=6). Secondly, the authors have not demonstrated that this morphology based system represents an independent prognosticator distinct from size or stage.

Reviewer #2: Thank you for asking me to review this paper by Noguchi and colleagues. The authors have reviewed 42 cases of IPNB (32 cases available for path review). The have proposed a new morphological classification and examined its clinical impact.

Overall, I found this paper to be interesting. The morphological classification is of value, although I suspect it will need a far larger study for this to be taken up into practice. It is very difficult to conclude much from a 42 patient study. I have made recommendations for improvement:

-How did you identify cases of IPNB? Was this from existing pathology records? From a database? By re-reviewing all existing cases of biliary tract tumors? How did you identify patients with IPNB prior to 2010, prior to the WHO definition being published? Did you review pathological specimens retrospectively? Did you search your institutional pathological database with keywords? This should be explained fully.

-In the methods, it would be best not to repeat definitions, but rather to present methods by which you went about determining whether a patient had IPNB and its assigned WHO subtype. This is not described and certainly should be. More importantly, was this done retrospectively by re-reviewing all identified cases? How many pathologists re-reviewed the slides?

-Please remove all data from the methods. (e.g. paragraph presenting typical cases of sub-types of IPNB, with number of cases, this belongs in the results)

-Please remove figure captions from the main manuscript, as it is very hard to read through properly.

-I would suggest removing p-values from table 1, as the number of patients is very small and the extreme number of hypothesis testing is not useful and subject to chance findings.

-What drives the differences in postoperative oncologic outcomes between branch-, main-, and mixed-type lesions? Is it simply a matter of complete surgical resection? If so, there really is not much of a difference, other than mixed lesions being more diffuse and difficult to resect fully.

-Recurrence/oncologic outcomes should be limited to invasive cancer cases of course.

-It is not clear to me how surgeons should use this classification? Does this influence the decision to resect patients?

-There are far too many figures for the sample size.

6. PLOS authors have the option to publish the peer review history of their article (what does this mean? ). If published, this will include your full peer review and any attached files.

**Do you want your identity to be public for this peer review?** For information about this choice, including consent withdrawal, please see our Privacy Policy .

Reviewer #1: No

Reviewer #2: **Yes: ** Guillaume Martel

---

## [Author Response · Author response to Decision Letter 0]

2 May 2025

Reviewer #1

General Comment:

The manuscript “A Recent Update on the Morphological Classification of Intraductal Papillary Neoplasm of the Bile Duct: Correlation with Postoperative Prognosis and Pathological Features” by Noguchi et al. examines 42 IPNBs and classifies them using a morphological system derived through radiologic characterization of IPNBs as branched type, main duct type, or mixed. They also classified them using the Japanese-Korean / WHO system as type 1 or type 2 based on histopathology. The authors evaluated postoperative survival based on several clinical-pathologic parameters, including by their proposed morphologic system. They found that mixed type IPNBs had worse survival compared to the others.

This paper represents an update on an earlier publication by this institution from 2013 and it includes more cases now. However, some fundamental flaws remain that make me skeptical of the validity of their proposed classification.

Our response to the general comment:

We sincerely appreciate your thoughtful and thorough evaluation of our manuscript. In this revised version, we have carefully addressed each of your comments and provided additional clarification to support the validity of our proposed morphological classification. We hope that our responses and revisions adequately address your concerns and improve the quality of the manuscript.

Comment 1: The mixed type only had 6 cases, a very small sample size that barely resulted in statistical differences on survival analysis. With so few cases, it is impossible to do meaningful multivariate regressions. My concern is that the mixed type may simply represent a more advanced stage disease, whereby tumor has expanded beyond its main duct or branch duct origin to involve adjacent regions. Although size or span can be difficult to measure in these types of lesions, I wonder if these are simply “larger” or at least more advanced stage tumors. In other words, I’m skeptical that the mixed morphology is an independent prognosticator that couldn’t be supplanted by other related, existing parameters such as size or stage.

Comment 2: There is some overlap between the intrahepatic versus extrahepatic anatomic classification as well as the type 1 and type 2 pathologic classifications with the proposed morphologic classification. Kim et al (reference 12) showed that intrahepatic versus extrahepatic location did not correlate with survival in a larger sample size. The authors argue that their morphologic system does correlate with survival due to more extensive spread in the mixed type. Couldn’t a similar classification be made for “mixed intrahepatic and extrahepatic involvement” instead? Do mixed intra and extra-hepatic tumours have different prognosis from those only involving intra or only involving extra?

Our response to the comment 1 and 2:

Your insightful comment highlighted a fundamental concern: the possibility that the poor prognosis observed in the mixed-type IPNB may simply reflect the adverse outcomes associated with larger or more extensively spreading tumors. If so, the need for a new morphological classification would be questionable. To address this concern appropriately, a comparison between our morphological classification and established indicators of tumor progression—such as tumor size or staging—would be required, ideally through multivariate analysis. However, as you also pointed out, the small sample size—the greatest limitation of our study—does not tolerate this statistical approach.

To address this issue despite the limited sample size, we conducted survival analyses comparing our morphological classification with indicators of tumor extent and progression. While indirect, we believe this clearly defined approach yields meaningful insights. First, we compared survival outcomes according to anatomical spread—limited to intrahepatic ducts, extrahepatic ducts, or involving both (Figure 2B). Survival did not significantly differ based on anatomical extent alone, suggesting that our morphological classification captures biological characteristics beyond tumor size or extent. Although the underlying factors remain unclear, the observed association with histopathological subtypes may be relevant. At the very least, morphological classification appears to be distinguishable from tumor extent. Second, to evaluate the impact of tumor progression, we focused on patients with invasive IPNB to enhance the validity of comparisons. Invasive IPNBs are treated as cholangiocarcinomas, and staging differs depending on tumor location—intrahepatic, perihilar, or distal bile duct—each with its own criteria. As you correctly noted, stage-based comparisons would be ideal, but due to differing staging systems, meaningful integration was not feasible. Instead, we used two common indicators of progression in bile duct cancers—depth of invasion and lymph node metastasis—to assess their association with prognosis. Both showed stronger statistical associations with prognosis than our morphological classification. However, these indicators are only available postoperatively, whereas our morphological classification can be assessed preoperatively, thus offering a distinct clinical advantage.

Taken together, our morphological classification may offer prognostic value beyond simply reflecting tumor size and its extent, and provide distinct clinical utility from tumor progression indicators by enabling preoperative prediction. The revisions are listed below.

Revisions to Figures:

• In Figure 2, DSS according to tumor extent was added. We also clarified that this analysis included all IPNB patients, distinguishing it from the survival analysis of patients with invasive IPNB shown in Figure 3. The legend for Figure 2 has been revised accordingly.

• In Figure 3, to evaluate the impact of tumor progression, we focused on patients with invasive IPNB in survival analysis according to pathological tumor progression factors. The legend for Figure 3 has been revised accordingly.

Revisions to manuscript:

• On Page 13: In the Results section “Postoperative survival by morphology and lesion extent in all IPNB cases”, we added the results of DSS by tumor extent as follows: As shown in Figure 2B, DSS did not show significant differences among the three groups based on tumor lesion extent (p=0.167 between intrahepatic and extrahepatic bile duct lesions, p=0.222 between intrahepatic and both lesions).

• On Page 14: In the Results section, we added a new subsection titled “Postoperative Survival by Pathological Tumor Progression Factors in IPNB with Associated Invasive Carcinoma,” which includes the following content: In thirty-one patients with IPNB with associated invasive carcinoma, DSS was compared based on pathological tumor progression factors. All patients with tumors confined to the mucosal (M) or fibromuscular (FM) layers survived without disease-specific death for 5 years after surgery. In contrast, patients with tumor invasion beyond the bile duct wall (serosal exposure [SE] or serosal invasion [SI]) had 1-, 3-, and 5-year DSS rates of 80%, 0%, and 0%, respectively, which were significantly lower (p < 0.001; Figure 3A). Patients with tumor extension to the subserosal (SS) or serosal (S) layers also had lower DSS compared to those with M or FM invasion (p = 0.037), although the difference became more evident after 5 years postoperatively. In patients with lymph node metastasis, the 1-, 3-, and 5-year DSS were significantly lower compared to those without lymph node metastasis. The DSS rates were 96%, 91%, and 91% versus 100%, 67%, and 67%, respectively (p=0.021, Figure 3B). In patients with residual tumor (R1 or R2), the 1-, 3-, and 5-year DSS were 86%, 64%, and 64%, respectively. Although these rates were lower than those in patients who underwent curative resection, the difference was not statistically significant (p=0.119, Figure 3C).

• On Page 16, Line 288: In the Discussion section, we emphasized that the morphological classification has significance beyond merely reflecting tumor extent, and that it offers clinical utility by being assessable preoperatively—unlike pathological factors that are strongly associated with prognosis but only available postoperatively. The revised sentences are as follows: However, lesion extent alone does not appear to fully account for the poor prognosis associated with the mixed type. As shown in Figure 2B, there were no significant differences in DSS among patients with intrahepatic, extrahepatic, or both intra- and extrahepatic tumor involvement. This suggests that our classification reflects not only the anatomical extent of the tumor but also underlying biological characteristics—for example, its association with pathological subtypes. We next examined the relationship between prognosis and cancer staging. In invasive IPNB, tumor stage is determined according to different classification systems based on tumor location—namely, intrahepatic, perihilar, or distal bile duct cancer—making direct comparison across cases impractical. Instead, we focused on two common oncologic factors: depth of invasion and lymph node metastasis, which can serve as surrogates for pathological staging. As shown in Figure 3, both were strongly associated with poor prognosis and, in terms of p-values, were even more predictive than morphological subtype. However, as these factors are only assessable postoperatively, our morphological classification offers a major advantage by enabling preoperative prognostic assessment and supporting treatment strategy.

Comment 3: The authors posit that the branch duct tumors are originating from peri-biliary glands, but since all the branch ducts tumors were intrahepatic, couldn’t they just be from smaller branches of intrahepatic ducts, rather than peri-biliary glands?

Our response to the comment 3: Thank you for your insightful comment. We revisited the literature regarding the histogenesis of IPNB and found that its origin is considered to be from both the peribiliary glands (PBGs) and the biliary epithelium (Ref#1). Since PBGs are distributed throughout both the intrahepatic and extrahepatic bile ducts, IPNBs originating from PBGs may potentially exhibit any of the morphological subtypes—branched, main duct, or mixed (Ref#2, #3). The same applies to lesions arising from the biliary epithelium. Therefore, although the precise cellular and tissue origins of IPNB remain to be unveiled, the idea that PBGs origin strictly corresponds to the branched type appears overly simplistic.

As you suggested, IPNBs derived from small bile ducts beyond the segmental ducts are likely to arise intrahepatically and to exhibit a branched-type morphology. We have added your consideration to the Discussion section and clarified that IPNB may arise from two potential origins as follows on Page 15, Line 259: Previous morphological classifications of IPNB considered the lesion location (intrahepatic vs. extrahepatic) as a key factor (8,10,12). In contrast, our new morphological criteria do not take lesion location into account. This is based on the understanding that IPNB can arise not only from biliary epithelial cells but also from peribiliary glands (PBGs) (20)—accessory glands distributed along both intrahepatic and extrahepatic bile ducts, which contribute to secretion and epithelial repair (18,21). Lesions derived from biliary epithelium, when arising in small bile ducts (after the segmental branches), typically may present as the branched type and be in the intrahepatic bile ducts. In contrast, PBG-derived lesions may occur irrespective of intrahepatic or extrahepatic bile duct, as PBGs are present throughout the biliary tract.

References:

#1 Nakagawa H, Hayata Y, Yamada T, Kawamura S, Suzuki N, Koike K. Peribiliary Glands as the Cellular Origin of Biliary Tract Cancer. Int J Mol Sci. 2018;19: 1745. doi:10.3390/ijms19061745

#2 Miyata T, Uesaka K, Nakanuma Y. Cystic and Papillary Neoplasm at the Hepatic Hilum Possibly Originating in the Peribiliary Glands. Case Rep Pathol. 2016;2016: 9130754. doi:10.1155/2016/9130754

#3 Uchida T, Yamamoto Y, Ito T, Okamura Y, Sugiura T, Uesaka K, et al. Cystic micropapillary neoplasm of peribiliary glands with concomitant perihilar cholangiocarcinoma. World J Gastroenterol. 2016;22: 2391–2397. doi:10.3748/wjg.v22.i7.2391)

Comment 4: With respect to survival analysis, disease specific survival is defined as per usual. However, the most common definition of “recurrence free survival” includes all deaths, but the authors chose to use a “recurrence-free disease-specific survival”, which is not a common oncologic outcome measure. I wonder if this very specific definition was chosen preferentially after multiple survival parameters were analyzed.

Our response to the comment 4: We sincerely apologize for the confusion caused. We would like to clarify that the recurrence-free survival (RFS) used in our analysis were not arbitrarily selected. Given that our study cohort includes patients treated as early as 1976, many patients have already died from non-disease-specific causes. Combined with the small sample size, we considered disease-specific mortality to be a more appropriate foundation for accurate time-to-event analysis related to tumor factors. Accordingly, we used disease-specific survival (DSS) as our primary survival endpoint and employed DSS + recurrence as the event definition for RFS.

However, as you rightly pointed out, this approach carries the risk of confusion, and since our intent was to assess recurrence alone, we decided to revise the analysis and simplify the content. Specifically, we now present cumulative recurrence rates instead of RFS. As also suggested by Reviewer #2, recurrence analysis should be limited to cases of invasive carcinoma. Therefore, we evaluated the relationship between morphological subtypes and cumulative recurrence rates in patients with IPNB with associated invasive carcinoma. The results have been added to Figure 4 and described in the Results section on Page 14, Line 242 as follows: Postoperative recurrence by morphology in IPNB with associated invasive carcinoma. The association between postoperative recurrence and morphological subtypes was evaluated in patients with invasive IPNB. No recurrence was observed in the branched type. In contrast, the mixed type showed a high recurrence incidence, with 1-, 3-, and 5-year cumulative recurrence rates of 25%, 25%, and 62%, respectively (p=0.074 vs. branched type, Figure 4). In the main duct type, the recurrence risk was intermediate, with 1-, 3-, and 5-year cumulative recurrence rates of 6%, 6%, and 13%, respectively, and no significant difference compared to the branched type (p=0.246).

Comment 5: Although the authors argue that the more aggressive nature of the mixed type could guide surgical management, the bulk of surgical decision making is still based on anatomical location, which dictates whether hepatectomy, bile duct resection, or pancreaticoduodenectomy is performed. In summary, although the finding that a mixed main/branched duct morphology potentially signals worse prognosis, this study has two main limitations. One is that the sample size for this category is very small (n=6). Secondly, the authors have not demonstrated that this morphology based system represents an independent prognosticator distinct from size or stage.

Our response to the comment 5: We sincerely appreciate your insightful and constructive feedback. As you rightly pointed out, the surgical approach for IPNB largely depends on the anatomical location of the lesion. In this context, we believe our proposed morphological classification provides clinical value in the determination of appropriate resection extent. For example, mixed-type IPNBs may require wider resections, while branched-type lesions could allow for more limited, minimally invasive approaches (laparoscopic or robotic). In cases where curative resection necessitates highly invasive procedures such as hepatopancreatoduodenectomy, this classification may support the rationale for surgical aggressiveness. In addition to these supportive roles, given the association between morphology and postoperative recurrence (Figure 4), our classification may contribute to postoperative treatmen

---

## [Editor Report · Decision Letter 1]

7 May 2025

A Recent Update on the Morphological Classification of Intraductal Papillary Neoplasm of the Bile Duct: Correlation with Postoperative Prognosis and Pathological Features

PONE-D-25-06283R1

Dear Dr. Kuriyama,

We’re pleased to inform you that your manuscript has been judged scientifically suitable for publication and will be formally accepted for publication once it meets all outstanding technical requirements.

Kind regards,

Sean Bennett, MD, MSc

Academic Editor

PLOS ONE

Additional Editor Comments (optional):

Thank you for making the necessary revisions and congratulations on an excellent paper!
---

## [Editor Report · Acceptance letter]

PONE-D-25-06283R1

PLOS ONE

Dear Dr. Kuriyama,

I'm pleased to inform you that your manuscript has been deemed suitable for publication in PLOS ONE. Congratulations! Your manuscript is now being handed over to our production team.

Kind regards,

on behalf of

Dr. Sean Bennett

Academic Editor

PLOS ONE